

# The Flying Laboratory FLab: Development and application of a UAS to measure aerosol particles and trace gases in the lower troposphere

Lasse Moormann[1,a], Thomas Böttger[1], Philipp Schuhmann[1], Luis Valero[2,3], Friederike Fachinger[1,a], Frank Drewnick[1,a]

[1] Particle Chemistry Department, Max Planck Institute for Chemistry, 55128 Mainz, Germany
[2] Institute for Atmospheric Physics, Johannes Gutenberg-University, 55128 Mainz, Germany
[3] Institute of Applied Geosciences, Darmstadt University of Technology, 64287 Darmstadt, Germany
[a] now at: Multiphase Chemistry Department, Max Planck Institute for Chemistry, 55128 Mainz, Germany

*Correspondence to*: Frank Drewnick (frank.drewnick@mpic.de)

**Abstract**

Unmanned aircraft systems (UAS) are gradually being established in environmental research to study boundary layer conditions and phenomena *in situ*; however, due to payload limitations, UAS can typically measure only a limited number of atmospheric variables simultaneously. Here we present the Flying Laboratory (FLab), a hexacopter equipped with six instruments to measure aerosol particles (particle number concentration and size distribution; $PM_{1/2.5}$ and black carbon mass concentration), trace gases ($CO_2$, $O_3$), and meteorological variables (temperature, relative humidity, pressure, wind) in the lower troposphere in real time and with high temporal resolution. The instrumentation has been selected to provide an overview of relevant variables in urban and semi-urban environments and especially in the vicinity of aerosol sources. This paper describes the development of the technical setup of the Flying Laboratory, the characterization of the measurements with respect to horizontal and vertical motion of the UAS, and the optimization of measurement flight patterns. During two field experiments, FLab was applied to bridge the gap between ground-based and aircraft-based profiling measurements and to perform hourly vertical profiling flights up to 300 m above a ground-based reference station for eight hours. These applications demonstrate the capability of FLab to capture the evolution of the lower convective boundary layer during the day and the vertical particle transport in the afternoon up to 200 m above ground.

## 1 Introduction

Atmospheric aerosol particles have a profound impact on air quality, human health, and global climate (Li et al., 2017). However, the tropospheric distribution of aerosol particles is far from being homogeneous, with strong gradients in aerosol



and trace gas concentrations, especially in the vicinity of sources. The situation is further complicated by the dynamics of the planetary boundary layer (PBL), which affects the transport of aerosol particles and trace gases and thus their distribution and concentrations (Stull, 1988). Therefore, to fully capture and characterize an emission plume, it must be analyzed in all three spatial dimensions, especially also vertically. However, even in the absence of strong sources, PBL dynamics can lead to strong

inhomogeneities of pollutant distributions, requiring a comprehensive, more than one-dimensional approach for their characterization.

For a detailed characterization of the aerosol and trace gas concentrations within the PBL, remote and *in situ* methods can be used. Remote sensing methods include the use of lidar and radar, which are capable of providing vertically resolved information on, for example, aerosol backscatter. They can be deployed both stationary (ground-based) and on mobile

platforms such as aircraft or satellites. While stationary deployment provides detailed information with high time resolution for a single location, the latter allows a broader view of the troposphere by capturing larger scale atmospheric phenomena (Hindman et al., 1984; Kotthaus et al., 2023), at the cost of a more limited temporal resolution. However, all of these remote sensing methods rely on the propagation of electromagnetic waves, which limits the observable variables to properties related to the scattering and absorption of these waves (Si et al., 2021).

In contrast, *in situ* methods involve direct measurements at the location of interest. Ground-based stationary measurements with high quality instrumentation can provide detailed information about the temporal evolution of, e.g., pollutant concentrations (Drewnick et al., 2012; Schrod et al., 2020) with few legal and instrumental restrictions, but only at a single location and without information about the horizontal or especially the vertical distribution. This can be overcome by the use of airborne research platforms such as balloons or aircraft, which can be equipped with *in situ* measurement instruments and

perform kilometer-scale flights up deep into the stratosphere (Mahnke et al., 2021; Ouchi et al., 2019). However, the lower boundary layer in particular is only limitedly accessible by these means. Unmanned aircraft systems (UAS), with their ability to ascend from the ground to the upper troposphere, have filled this gap in recent years and can complement networks of ground-based measurement stations for a three-dimensional view of the lower boundary layer (Falco et al., 2021; Rabins et al., 2023). In addition to their affordability, an advantage of UAS is their precise maneuverability, which allows them to

perform vertical profiling with minimal horizontal deviation or defined mapping of targeted areas or volumes.

Depending on the scientific objective, fixed- or rotary-wing UAS are used. Fixed-wing UAS can be used to probe larger areas with flight distances and altitudes of up to several kilometers for several hours, but require runways or landing nets and experienced pilots (Reuder et al., 2012; Roberts et al., 2008). In contrast, rotary-wing UAS are more limited in flight duration, but can be controlled by less experienced personnel and allow for even more flight patterns, such as hovering. Because of these

advantages, hexacopter and other rotary-wing UAS have recently found many applications in PBL research and are gradually becoming established in tropospheric research (Hervo et al., 2023; Sziroczak et al., 2022).

Applications include the investigation of turbulent fluxes, boundary layer stability, or condensation phenomena (Adkins, 2020; Hamilton et al., 2022), but also wind field measurements at wind turbines (Adkins and Sescu, 2017; Li et al., 2022), or the investigation of wind turbine propeller efficiency degradation due to icing (Gao et al., 2021). Aerosol particle and trace gas



sensors are commonly deployed on UAS for the investigation of particle transport, photochemical processes, or tropospheric dynamics (Miller et al., 2024; Roberts et al., 2008). In addition to such studies in typically unpolluted environments, local emission sources are also often investigated, whether natural sources such as coastal wave breaking or volcanic activity (Brady et al., 2016; Galle et al., 2021; Lappin et al., 2023; Radtke et al., 2023) or anthropogenic sources such as landfills, refinery platforms, agricultural farms, or other industrial sites (Allen et al., 2019; Andersen et al., 2023; Bonne et al., 2024; Castro

Gamez et al., 2019; Gålfalk et al., 2021; Golston et al., 2018).

A critical limitation for such studies, however, is the limited payload of the UAS, which allows either very limited information with a single or few high quality instruments (Womack et al., 2022), or the use of multiple lightweight but lower quality sensors, which can provide data of low and frequently questionable quality and often require complex correction procedures (Schuldt et al., 2023). Offline analyses of aerosol filter samples or, for the gas phase, desorption tubes, Tedlar bags, or Aircore

tubes have also been applied (Andersen et al., 2023; Liang and Shen, 2023; Niedek et al., 2023; Zhu et al., 2024), but all have the disadvantage of being both time consuming and limited to the investigation of only a few variables, typically at a very limited number of locations during a single flight. A first approach to overcome this limitation was presented by Brus et al. (2021a) and Pohorsky et al. (2024), who developed different modular systems for aerosol particle and gas phase analysis which were mounted on a rotary-wing UAS or a tethered helikite, respectively. While this is a step towards a more comprehensive

investigation of the troposphere, the simultaneous measurement of a variety of aerosol, trace gas, and meteorological variables is crucial for understanding microphysical processes, especially in the PBL. However, to our knowledge, no UAS capable of providing such a broad overview has been presented in the literature due to the aforementioned obstacles.

Here we present the development and application of the hexacopter-based Flying Laboratory (FLab), which can be used to study aerosol particles, trace gases, and meteorological variables simultaneously and in real time in the lower boundary layer.

The instrumentation was selected to cover relevant variables commonly used for air pollution studies in semi-urban and urban environments, especially in the vicinity of aerosol sources, and to mirror key variables simultaneously measured on-board the ground-based research platform MoLa (Mobile Laboratory, Drewnick et al. (2012)).

Section 2 describes the technical and instrumental setup of FLab, with emphasis on the arrangement of the instruments for undisturbed measurements and the electronic setup. In Section 3, we analyze and discuss the uncertainty of the measured

variables on board FLab and possible measurement biases due to the flight motion. Finally, Section 4 presents some exemplary research applications that illustrate the benefits of hourly vertical profiling measurements and demonstrate how UAS-based measurements can bridge the gap between ground-based and airborne measurements.

## 2 The Flying Laboratory FLab

### 2.1 Description of the FLab platform

The Flying Laboratory is based on a commercial DJI M600 hexacopter, which has a base weight of 9.1 kg (including batteries) and is designed to carry a payload of up to 6 kg. In its current configuration, FLab has a diameter of 1.58 m and a height of





1.86 m. The payload consists of instruments for the measurement of aerosol particles, trace gases, and meteorological variables, as well as of electronic infrastructure for power supply and data management. Most of the payload is installed in an aluminum rack mounted under the UAS to keep the center of gravity of the entire system below the UAS body (Fig. 1a). In addition, an

anemometer and an optical particle counter (see Sect. 2.2 for details), two instruments for which undisturbed advection of ambient air is critical, are located at a distance of 110 cm and 100 cm above the rotor plane, respectively. They are mounted on a frame consisting of two vertical carbon fiber tubes with custom 3D-printed organic polymer cross connectors to minimize vibration during operation at the lowest possible weight. In addition to these two instruments, a FLARM transponder (ATOM UAV, FLARM Technology AG), which is a traffic awareness and collision avoidance system for airborne objects and is quasi-

mandatory in the European Union for the authorization of flights above 120 m AGL (above ground level) (European parliament and council, 2019), is also mounted at an elevated position on the frame. This position was chosen to allow unobstructed radio emission from the device. For nighttime measurements, the frame can be equipped with an LED band for better visibility of the UAS.

The FLab power supply is divided into two completely independent units (Fig. 1b). The UAS, including propulsion, UAS

electronics and FLARM, is powered by six intelligent flight batteries (DJI TB47S, LiPo 6S, 99.9 Wh). The FLab payload, including instruments and data acquisition electronics, is powered by an on-board battery (type: GBA 2.0 Ah, Robert Bosch Power Tools GmbH), or alternatively, between flights, by the custom-built 18 V ground power unit, to which it can be switched without interruption. This allows uninterrupted operation of the payload electronics when the on-board battery is changed. Both power supplies provide 18 V power to multiple DC/DC converters that generate voltages from 5 V to 18 V to supply all

instruments as required.

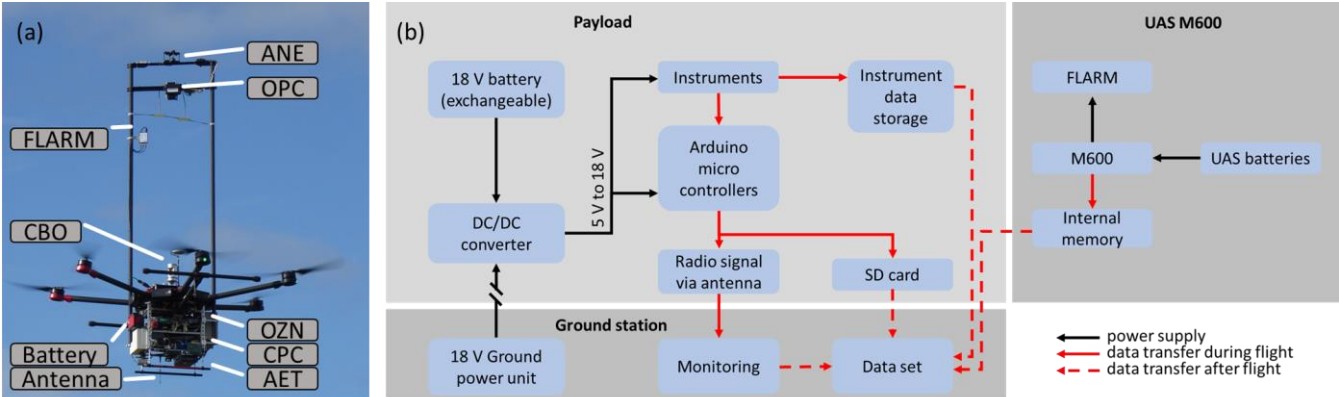

**Figure 1: Photo of FLab with the positions of the measurement instruments and the components of the electronic infrastructure indicated; see Table 1 for an explanation of the acronyms (a). Schematic layout of the FLab electronics, showing the power supply**

**infrastructure (black arrows) and data management (red arrows) for different parts of FLab (b).**





The complete FLab dataset of an aerial mission includes instrument data and system data from the M600 UAS (Fig. 1b). The aerosol instruments transmit measurement data to Arduino Mega microcontrollers (ATmega 2560). Due to the low processing power of the Arduino Mega, three microcontrollers are required in the FLab to receive pre-processed data and to process the

partially large output strings. An Arduino Uno (ATmega, 328P) is used to store the processed instrument data on a common SD card for all instruments. The data stored on the on-board SD card are simultaneously transmitted to the ground station via a serial transceiver module (HM-TRP-RS232 series 100 mW, Shenzhen Hope Microelectronics Co., Ltd., China) and an antenna (DeLOCK ISM 433 MHz, type 88877, Tragant GmbH, Germany) located under the payload rack. At the ground station, the data is recorded along with the current ground station time. The transmitted data can be evaluated in real time by

a crew member who can alert the pilot about any unexpected weather conditions or plumes. The M600 UAS stores various technical variables such as orientation, velocity, GPS location, propeller rotation rate, etc. on an internal storage device. However, this data is not accessible to the crew during operation and must be downloaded after the flight for further processing (Fig. 1b). Some instruments (ozone monitor and aethalometer, see Table 1) also store their data in independent internal memories.

This results in four data sets: the M600 UAS data set, two instrument data sets (stored individually and on the common SD card), and the data received from the ground station. All of these are merged into a single common data set by aligning them to the reference time stamps, namely those of the ground station. To do this, they are correlated with varying time shifts to the corresponding ground reference time series until the Pearson correlation coefficient is at its maximum (typically very close to 1). To align the M600 UAS data, the altitude data from the M600 UAS are correlated with the pressure data recorded by the

FLab anemometer to correct for any time shift between the data sets.

## 2.2. Instrumentation

The instruments installed on board FLab were selected to cover the most relevant measurable quantities for a robust investigation of anthropogenic emission plumes and boundary layer dynamics. Therefore, we chose instruments for the following variables: particle number concentration (particle diameter $d_p > 10$ nm), particle size distribution (optical diameter:

0.35 µm $< d_{opt} < 40$ µm; also used to determine particulate mass concentrations up to 1 µm and 2.5 µm particle diameter, i.e., $PM_1$ and $PM_{2.5}$), black carbon mass concentration, $O_3$ and $CO_2$ volume mixing ratios, and meteorological variables (wind direction and speed, temperature, pressure, relative humidity). The DJI M600 can carry a maximum payload of 6 kg, which must include the instruments, but also the power supply, data acquisition electronics, and mechanical installation. Considering this limitation, the instruments had to combine aspects such as low weight and compact size with high data quality and

robustness as well as high time resolution of the measurements, resulting in the FLab setup as presented in Table 1.

Several instruments have been modified to meet the requirements of on-board FLab operation. All instruments with RS232 interface required modifications to convert the serial output to 5V TTL (Transistor-Transistor Logic) level, which is read by the microcontroller. From the CBO, CPC, and OZN, the housing was removed from the factory version of the instrument to reduce weight, while the AET and OPC have increased weight due to modifications to their inlet system (see Table 1). On the





bottom side of the ANE, 1 cm² of the housing and a membrane (which protects the ANE electronics from condensation during extreme humidity changes) were removed to reduce the humidity sensor measurement delay. Without the membrane, the ANE relative humidity and temperature data were in best agreement with the reference instrument; therefore, we relied on the ANE rather than the CBO or OPC (see Table 1) for both variables in all further analyses. The CBO required additional adjustments for in-flight operation: a filter cage was removed from the optics to allow faster adjustment of the measurement volume to

ambient $CO_2$ levels, and the in-flight data from this instrument is pressure and temperature corrected using formulae provided by the manufacturer. The CPC and OZN did not require adjustments for in-flight operation other than calibration. For the AET, a laminar inlet flow of 170 cm³ min⁻¹ is passed through a 30 mm long inlet tube with an inner diameter of 3.3 mm. The AET measures in dual-spot or single-spot mode (referring to the number of spots analyzed simultaneously). The dual-spot mode compensates for mass loading effects on the filter on which black carbon is collected, while the single-spot mode has a 30%

higher sample flow rate and therefore reduced noise (Drinovec et al., 2015).

**Table 1: FLab instrumentation.**

| Acronym | Instrument | Measured quantity | Time resolution | Installation weight (previous if modified) |
|---------|-----------|-------------------|-----------------|--------------------------------------------|
| AET [a] | Aethalometer | Black carbon mass concentration | 2 s | 425 g (420 g) |
| ANE [b] | Anemometer | Horizontal wind speed and direction; temperature; relative humidity; pressure | 1 s* | 60 g |
| CBO [c] | Carbon dioxide monitor | Mixing ratio of $CO_2$; temperature | 2 s** | 320 g (360 g) |
| CPC [d] | Condensation particle counter | particle number concentration | 1 s | 810 g (1.7 kg) |
| OPC [e] | Optical particle counter | particle size distribution based on optical diameter; temperature; relative humidity | 1 s | 240 g*** (105 g) |
| OZN [f] | Ozone monitor | Mixing ratio of $O_3$ | 2 s | 1.16 kg (2.6 kg) |
| DJI [g] | UAS: DJI Matrice 600 (M600) | 3D orientation; 3D flight velocity; GPS position; wind speed and direction; altitude based on pressure level and GPS; propeller rotation rate; various internal data | ≤1 s | 9.1 kg (with battery set) |

[a] microAeth® MA200, AethLabs, USA. [b] TriSonica™ Mini, Anemoment LLC, USA. [c] CARBOCAP® Carbon Dioxide Probe GMP343, Vaisala Ojy, Finland. [d] Condensation Particle Counter Model 3007, TSI, Inc., USA. [e] OPC-N3, Alphasense AMETEK®, United Kingdom. [f] Model 205 Dual Beam Ozone Monitor, 2B Technologies, Inc., USA. [g] Matrice 600, SZ DJI Technology Co., Ltd., China.
*internal sampling frequency of 40 Hz, but output frequency of 1 Hz. **data output is at 1 Hz, but new data points are measured with 0.5 Hz. ***final installation weight including external pump.



We tested both modes for our application (see Sect. 4). Transport losses and sampling delays can be neglected for all
instruments as no long sampling tubes were used and all corresponding sampling delays are well below the 1 s temporal
resolution of the data acquisition.

The installation positions of the instruments were based on the individual requirements of the respective measurements. For
the anemometer and the OPC (which also measures particles with $d_{opt} > 1 \ \mu m$), a mounting position was chosen that is as little
affected as possible by the downwash from the UAS propellers (see Sect. 2.2.1 and 2.2.2). The measurements with the gas
sensors and of the submicron particles (CPC and AET) are not strongly affected by the wind from the propellers. Therefore,
these instruments, as well as the power supply and data acquisition electronics, were installed below the body of the UAS (the
CBO directly on top of the body) to counterbalance the anemometer/OPC frame and to keep the center of gravity of the FLab
as low and centered as possible.

### 2.2.1 Wind measurements on board FLab

Wind speed and direction measurements aboard a UAS are always a measurement of the wind relative to the platform. To
obtain absolute values, this data must be combined with GPS velocity data from the UAS. There are several ways to determine
the relative wind on board a multicopter, including installing anemometers on the UAS for a direct measurement, or estimating
wind speed based on the pitch angle and power consumption of the UAS propellers. The pitch angle method is accurate, but it
requires extensive study of the flight behavior of the UAS used, which can change with changes in its weight or center of mass
(Wildmann and Wetz, 2022). Thus, while the M600 UAS control software provides absolute wind velocity and direction data
based on this method (Mathes, 2023), these values must be used with caution, as the flight behavior of the UAS may have
changed due to the attachment of the payload, resulting in potentially biased calculations. Therefore, an anemometer was
installed to directly measure the relative wind.

Multicopters perturb the surrounding wind field by aspiration of air from above and sidelong the rotors, creating a strong
downwash below the UAS. External anemometers are typically installed on the aspiration side at some distance from the rotors
to minimize the air perturbations produced by the propellers (Adkins, 2020; Thielicke et al., 2021). The intensity and extent
of the perturbed wind field varies with the distance from the rotors and the number and size of the rotors. Therefore, the
perturbed wind field needs to be investigated individually for each UAS (Abichandani et al., 2020).



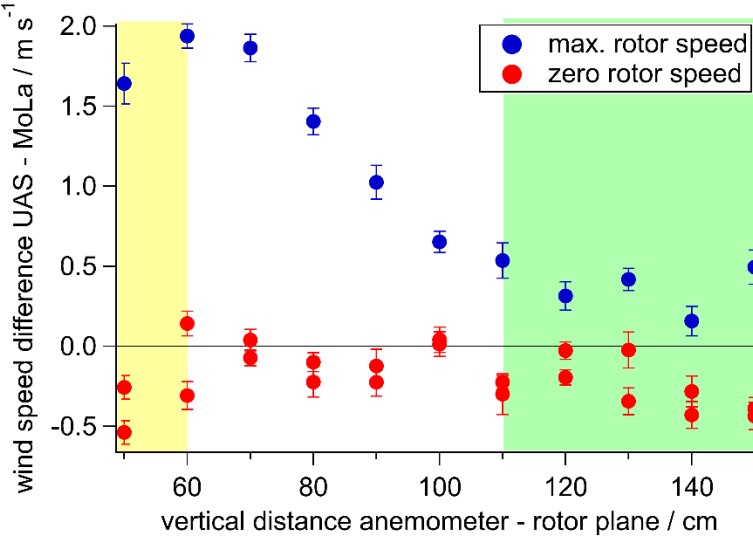

**Figure 2: Difference in wind velocities measured by MoLa and UAS anemometers at maximum (blue) and zero rotor speed (red), for different UAS anemometer mounting positions. The downwash-related bias was constant when the anemometer was mounted at least 110 cm above the rotor plane (green); the data are consistent with a dead volume below 60 cm (yellow). Error bars are calculated from the standard errors of the MoLa-/UAS-based measurements using Gaussian error propagation.**

In order to investigate the influence of the UAS propeller operation on the measured ambient wind speed at different mounting positions of the anemometer above the UAS propeller plane, experiments with a fixed UAS were performed. For this, it had to be taken into account that the near-ground operation of a multicopter results in a highly disturbed wind field around the aircraft. For a JF01-10 crop protection UAS of comparable size, it was found that the wind field perturbation was no longer disturbed by the ground when operating at an altitude of at least 3 m above the ground (Zheng et al., 2018). Therefore, we mounted the M600 UAS on a forklift at a height of about 5 m above ground and placed it 5 m away from a reference 2D ultrasonic anemometer (WXT520, Vaisala Ojy) mounted on the Mobile Laboratory (MoLa, Drewnick et al., 2012) at the same height. The UAS anemometer was installed at heights ranging from 50 cm to 150 cm above the UAS propeller plane and was always kept at the same height as the reference anemometer. Three measurements were performed under ambient conditions for 2 minutes each: twice with the propellers turned off and once at the maximum rotation rate. During the measurement period, the absolute wind speed measured with the reference anemometer ranged from 1.4 m s$^{-1}$ to 4.0 m s$^{-1}$, and the wind direction varied within a range of 132°. No disturbance of the wind measured by the reference anemometer was found when the UAS was turned on.

For each anemometer position, the difference between the averaged wind speeds measured on the UAS and on MoLa was calculated for both measurement situations, i.e., with the propellers operating and with the propellers turned off. No difference in the wind speeds indicates 100% agreement between the FLab and MoLa anemometers. The results without rotating propellers were used to ensure the comparability of the measurements with the respective instruments when the wind field was not disturbed by the UAS. Figure 2 shows that without rotating propellers there is reasonable agreement between the UAS





anemometer and the MoLa instrument for all anemometer positions (red markers). Under normal flight conditions, i.e., with the propellers rotating at full speed (blue markers), the largest difference between the UAS and the reference anemometers is found for the mounting position 60 cm above the propeller plane. At lower mounting heights, a decreasing perturbation of the wind field is observed, consistent with a dead volume close to the rotor plane, which has been proposed as a possible mounting position for hover flights (Li et al., 2023). Since horizontal movement of the UAS would drag the dead volume into an already disturbed regime, this mounting position is not suitable for general flight operations. For mounting positions more than 60 cm above the propeller plane, the wind difference between the FLab and MoLa decreases with increasing distance from the rotor plane and stabilizes at a distance of about 110 cm. Since the position of the anemometer should be as close as possible to the UAS body in order to minimize the weight of the tube for the mounting frame and the leverage in case of strong gusts, the ideal height for mounting the anemometer is 110 cm above the rotor plane. However, it cannot be ruled out that the anemometer may still be affected by wind distortion caused by the UAS in situations with a very high pitch angle, such as when the UAS is flying into strong winds. The anemometer wind data were transformed from the body-fixed reference system via yaw, pitch, and roll angles to the terrestrial reference system using 3D rotation matrices and converted to absolute (above ground) velocities using the flight velocity recorded by the UAS (Thielicke et al., 2021).

### 2.2.2 OPC sampling inlet optimization

The optical particle counter (OPC) measures particles larger than 350 nm in optical diameter. To minimize sampling bias of aerosol particles into the instrument due to the strong airflow caused by the propellers, we mounted the OPC approximately 100 cm above the rotor plane with the inlet pointing upwards and no inlet tube attached. In this mounting position, directly below the anemometer (see Sect. 2.2.1), the perturbation of the ambient wind field by the UAS propellers is very small, resulting in a minimal influence on the sampling efficiency. Since no inlet tube is used, transport losses to the instrument can be neglected in the very short and vertically oriented inlet of the OPC.

In the original configuration of the instrument, the sample flow of the OPC is generated by a low-power fan and is determined by the particle time-of-flight through the measurement volume (Alphasense, 2019). During the first test flights of FLab, we observed a strong dependence of the OPC sample flow rate on the UAS vertical velocity (Figure S1a, red markers). To reduce the influence of the vertical velocity of the UAS on the OPC sample flow, we installed plates above the inlet and below the outlet of the OPC to shield the sample flow from the vertical motion of the ambient air (Fig. S1b). The effect of the plates was evaluated for different plate distances from the OPC between 3 mm and 25 mm with test flights at different vertical velocities (Fig. S1a, blue markers).

The most stable setup in terms of flow dependence on vertical UAS velocity was found for the configuration with a distance of 10 mm from the front plate to the inlet and 9.5 mm from the outlet of the OPC to the outlet plate (Fig. S1a), but some dependence of the sample flow on the flight velocity remained. Other groups have demonstrated for a similar OPC setup that an external pump stabilizes the flow and allows for a more constant particle throughput without compromising instrument performance (Bezantakos and Biskos, 2022). To further stabilize the sample flow, we replaced the low-power fan with an



external pump (G 6/02 EB rotary vane pump, Metzger Technik GmbH) and sealed leaks in the OPC housing near the data
output cables that would otherwise reduce the sample flow (Fig. S1c). Figure S1a (black markers) confirms that the sample
flow in this setup appears to be independent of vertical velocity, but is slightly reduced compared to the fan-driven setup.
Previous studies for the similarly constructed OPC-N2 (Alphasense AMETEK®) found no dependence of the measurement
results on temperature > 5 °C and pressure > 700 mbar (Bezantakos et al., 2018), where our measurements also took place.

Therefore, no temperature or pressure correction was applied to the data, while hygroscopic particle growth was corrected
according to the Köhler theory using the meteorological data from the anemometer and an assumed Köhler κ of 0.3 for particles
in continental regions (Andreae and Rosenfeld, 2008).

## 3 Characterization of FLab performance

A series of dedicated experiments were performed with FLab to evaluate its performance in terms of time resolution and under

vertical and horizontal motion.

Ground-based measurements over an extended time interval (several days) were performed at the Max Planck Institute for
Chemistry in Mainz (semi-urban environment) to determine the uncertainty of the data from each instrument for different
averaging intervals and thus the achievable temporal and spatial resolution (Sect. 3.1).

In-flight experiments were performed at a rural site in a wine-growing area near Ockenheim, Rhineland-Palatinate, Germany

(Sects. 3.2 and 3.3). At this site, a rather homogeneously mixed aerosol is expected due to the distance to the residential areas
of Ockenheim (550 m to the east) and Kempten (1.1 km to the north) and to major roads, especially a highway at a distance of
900 m in the northwest-west direction (Fig. S2). Horizontal-only and vertical-only characterization flights were performed on
26 March and 10 April 2024, respectively, under sunny and cloudy conditions (Table 2). During the experiments, the weather
station and corresponding MoLa trace gas and aerosol measurements (Table S1) with a sampling height of 6 m AGL served

275    as reference for the FLab data.

As mentioned above, the measurement site for the in-flight characterization was chosen to be far enough away from any local
sources to allow experiments to be conducted under homogeneous conditions. Since the AET was designed specifically for
black carbon plume detection, this had the disadvantage that the measured black carbon mass concentrations were close to the
detection limit during all flights. AET determines the black carbon mass concentration from the attenuation at five different

280    wavelengths, with the lowest wavelength (370 nm) having the lowest noise level. Therefore, in the analysis (Sect. 3 and 4),
we only considered the black carbon concentration determined from the 370 nm measurement, noting that black carbon
concentrations are typically determined with wavelengths from the infrared spectrum (Pikridas et al., 2019).

Two different types of wind data were collected with FLab: in addition to the anemometer (ANE), the on-board computer of
the UAS collects wind data determined from the pitch angle and rotor speed (DJI; see also Sect. 2.2.1).

285





**Table 2: Flight patterns of the characterization flights.**

| Flight number | Date and local take-off time, weather condition | Flight duration | Flight pattern | Flight velocities* |
|---|---|---|---|---|
| F1 | 26 March 2024 at 15:34, sunny | 14 min 56 s | Horizontal (20 legs at 6 m AGL, ± 100 m from take-off site towards north/south) | 0 m s$^{-1}$, 2 m s$^{-1}$, 4 m s$^{-1}$, 6 m s$^{-1}$, 8 m s$^{-1}$, 10 m s$^{-1}$ |
| F2 | 26 March 2024 at 16:21, sunny | 14 min 18 s | Horizontal (19 legs at 6 m AGL, ± 100 m from take-off site towards north/south) | 0 m s$^{-1}$, 2 m s$^{-1}$, 4 m s$^{-1}$, 6 m s$^{-1}$, 8 m s$^{-1}$, 10 m s$^{-1}$, 15 m s$^{-1}$ |
| F3 | 26 March 2024 at 16:45, sunny | 14 min 37 s | Horizontal (19 legs at 6 m AGL, ± 100 m from take-off site towards north/south) | 0 m s$^{-1}$, 2 m s$^{-1}$, 4 m s$^{-1}$, 6 m s$^{-1}$, 8 m s$^{-1}$, 10 m s$^{-1}$, 15 m s$^{-1}$ |
| F4** | 10 April 2024 at 13:20, cloudy | 14 min 31 s | Vertical (7 legs from 6 m to 120 m AGL and back, hovering for 5 s in 6 m and 120 m AGL) | 0 m s$^{-1}$, 1 m s$^{-1}$, 2 m s$^{-1}$, 3 m s$^{-1}$ |
| F5 | 10 April 2024 at 13:42, cloudy | 14 min 18 s | Vertical (6 legs from 6 m to 120 m AGL and back, hovering for 5 s in 6 m and 120 m AGL) | 0 m s$^{-1}$, 1 m s$^{-1}$, 2 m s$^{-1}$, 3 m s$^{-1}$ |
| F6 | 10 April 2024 at 14:02, cloudy | 13 min 55 s | Vertical (6 legs from 6 m to 120 m AGL and back, hovering for 5 s in 6 m and 120 m AGL) | 0 m s$^{-1}$, 1 m s$^{-1}$, 2 m s$^{-1}$, 3 m s$^{-1}$ |

*0 m s$^{-1}$ correspond to hover flights at the turning points between the horizontal/vertical legs.
**the last ascent was performed up to 60 m AGL due to low battery capacity.

## 3.1 Instrumental time resolution and uncertainty

The operation of instruments onboard mobile platforms always requires a trade-off between extending the averaging times of the instruments to minimize the uncertainty of the data and reducing the averaging times to improve the temporal resolution and hence the spatial resolution of the data. This problem is aggravated in UAS measurements because the total sampling time during a single UAS flight is typically very limited by battery capacity (12-16 minutes in our case), and at the same time measurements over extended spatial distances with good spatial resolution are desired. In order to find a compromise that meets our measurement needs, we determined the dependence of the individual instrumental uncertainties on the duration of the averaging intervals in dedicated laboratory experiments.

In order to quantify the uncertainty for each variable as a function of sampling and averaging time, ground-based measurements of ambient air were performed with the FLab instruments for several hours up to several days (see description at beginning of Sect. 3). During this period, local emissions and diurnal cycles caused a changing intensity of the measured variables. By calculating the Allan variance (i.e., the variance of the difference of adjacent data points) as a function of averaging time, the statistical noise can be separated from the influence of temporal trends on the signal. Temporal trends dominate the average of data points for averaging times greater than the minimum of the Allan variance (Werle et al., 1993). Depending on the observed variable, we found the minimum Allan variance between 80 s and 100 s averaging time for our instruments (Fig. S3). Therefore, to estimate the statistical uncertainty of the measured variables, strictly speaking, only periods up to 100 s can be considered trendless. The raw data (with temporal resolution according to Table 1) were averaged in increments according to the respective sampling times of interest (ranging from 2 s to 100 s). The relative standard deviations (RSD) of the data



averaged in this way were calculated using a sliding window approach for a window size of 600 s. The window size was extended beyond the 100 s range (the maximum trendless averaging time according to the Allan variance analysis) in order to

be able to calculate relative uncertainties for averaging intervals up to 100 s with at least 6 values in each RSD calculation. Calculations with window sizes of 100 s and 600 s showed for averaging times up to 20 s that this increase in window size leads to small relative increases (0.1% to 13% of the respective values for the shorter window size) in the RSD obtained for most variables and instruments, with the exception of CPC (+40%) and pressure measurement (+95%), two variables that have low to extremely low relative uncertainties (see Fig. 3). Finally, from these RSDs, the median was calculated for each target

averaging time to eliminate outliers due to short-term events in the data and to minimize the influence of trends in the concentration time series.

Figure 3 shows that the RSD decreases with increasing averaging time for all instruments. The magnitude of the RSD is strongly dependent on the measured quantity and the individual instrument. If purely statistical effects cause the RSD, it should decrease with increasing averaging time $t$ with $t^{-0.5}$ (slope of the dotted line in Figure 3).

Instruments that measure particle mass concentrations, such as the AET and the OPC, where the output is based on the measurement of a relatively small number of particles per unit time, should follow this $t^{-0.5}$ statistical dependence. While this is approximately the case for the OPC, a steeper decrease in RSD was found for the AET for averaging times between 10 s and 100 s. We suspect that this is due to the internal processing of the raw data by this instrument and possibly to the fact that condensation and evaporation effects on the filters increasingly cancel each other out with longer averaging times. For

instruments where counting statistics are not critical to instrument performance, such as the CPC (where large numbers of particles are measured in short time intervals), the $O_3$ or $CO_2$ monitor (which measure a quasi-continuum of molecules), a smaller slope in the RSD-$t$ dependence is observed. This is due to the fact that for these instruments the precision is determined by a combination of other influences such as electronic noise or measurement cycles. For the $CO_2$ instrument and the CPC, averaging times above 40 s and 80 s, respectively, do not lead to a further reduction in uncertainty.

To calculate the statistical uncertainty as a function of the selected averaging time, we found that Hill equations were best suited to parameterize the results of our measurements. The Hill equations were fitted to the RSD of all studied variables for averaging times between 2 s and 100 s. The coefficients of the corresponding Hill equations are listed in Table S2.

From Figure 3 we found that averaging times of more than 10 s correspond to a statistical uncertainty of 0.1% to 20% for most variables and up to 100% for black carbon, while for 30 s averaging times 0.07% to 10% and 50%, respectively, were

determined. This means that for vertical profile measurements, a resolution of height bins in the order of a few tenths of meters can be achieved with reasonable measurement uncertainty, depending on the vertical flight velocity.





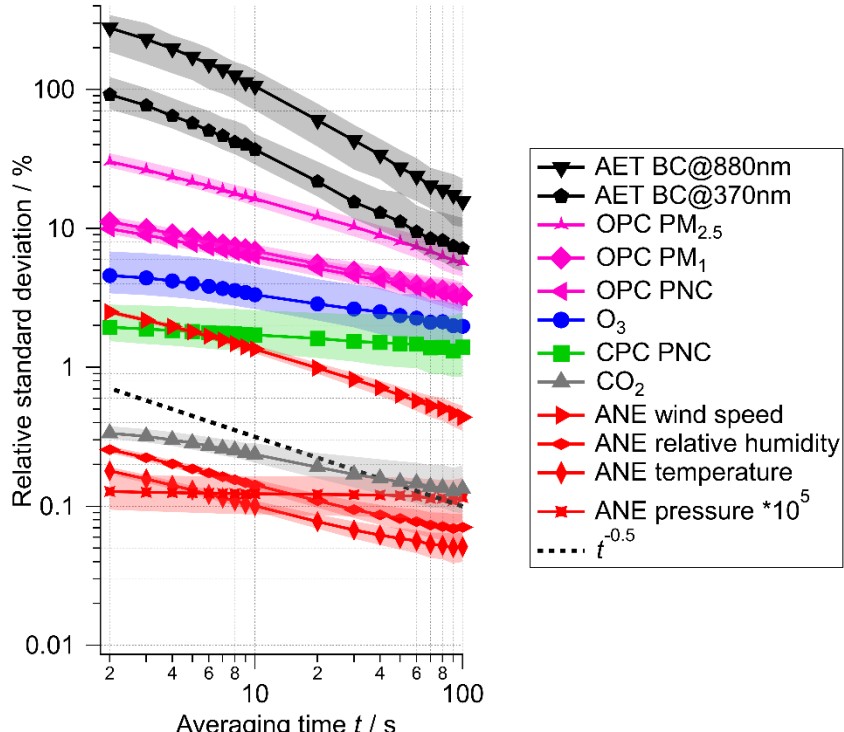

**Figure 3: Statistical uncertainty (relative standard deviation) of the measured quantities onboard FLab as a function of averaging time. Same colors correspond to the same instruments that recorded the data. The markers show the median over the time series and the shaded area shows the corresponding interquartile range.**

## 3.2 Influence of horizontal motion

In order to determine whether the horizontal motion of the UAS affects the measurement on board FLab, we performed three dedicated experiments with horizontal flights (F1 to F3, Table 2), all at 6 m AGL. Each flight consisted of several round trips from the start position to 100 m north, back past the start position until the UAS was 100 m south of the start point, and finally back to the initial start position. Each of these flight patterns was executed at a pre-programmed horizontal velocity, which was kept constant over the entire distance. During each flight, several patterns were completed at different velocities, as listed in Table 2.

As shown in Figure S4, during flight F1 at 15:44 LT (local time) a 40° change in wind direction occurred, which was associated with a change in several variables such as $CO_2$ volume mixing ratio, CPC PNC (particle number concentration), relative humidity and temperature. In contrast, conditions were stable during F2 and F3. Nevertheless, all flights are used to assess the effect of horizontal motion on instrument performance.

For each flight, the collected data from all instruments were binned in two different ways, after verifying that the flight direction did not affect the measured quantities. By binning the data for all horizontal velocities in 20-m increments as a function of the



horizontal flight distance from the start position it is possible to investigate potential small scale inhomogeneities at the measurement site. For this purpose, the mean and standard error of all data were calculated for the respective bins and demonstrate that within a range of 200 m gradients and structures of, e.g., temperature, wind speed and CPC PNC (Fig. S5) can be resolved. These structures may be due to the sloping topography or to local emission sources such as the MoLa exhaust, which polluted the flight path from -60 to 40 m in the case of easterly winds.

In order to investigate potential effects of flight velocity on the measurements, the acquired data need to be normalized with respect to their temporal and spatial variations. First, temporal variations of the FLab data were corrected using normalization factors derived from the 15 s rolling mean time series of the MoLa data, normalizing the UAS data to the first data point of each variable. Second, a correction for spatial structure as presented in Fig. S5 was applied by normalizing the UAS data to the 0 m mark of each flight for each variable. Finally, the mean of all flights at a given velocity was calculated for each variable. Combined uncertainties from temporal and spatial normalization, individual instruments, and flight-to-flight variability were estimated. Figures 4 and S6 show the resulting dependence of different variables on flight velocity and apparent wind speed, respectively. The apparent wind speed is calculated by adding the vectors of flight velocity and ambient (horizontal) wind.

Most variables are either not affected by high flight velocities, such as black carbon mass concentration, $CO_2$, $O_3$, and relative humidity, or, like the CPC, show an insignificant deviation of 1%, well within its uncertainty. However, the particle number concentration measured with the OPC decreases significantly at flight velocities > 6 m s$^{-1}$, consistent with apparent, though insignificant, trends in $PM_1$ and $PM_{2.5}$. Here, the vertical inlet may lead to inefficient sampling at high horizontal aspiration velocities, which could potentially be corrected if necessary, e.g., under very windy conditions (Brockmann, 2011),which is possible due to small trend uncertainties. Also, the measured pressure decreases with increasing flight velocity, probably due to the Bernoulli effect; the measurement at (nominally) 0 m s$^{-1}$ may be slightly affected by the apparent wind from the UAS rotation at the end of each flight lag, causing the slight pressure drop. The temperature in the ANE is calculated from the speed of sound and therefore relies on the measured pressure, causing a similar trend (Figure S6e); however, both trends are negligible compared to the measurement uncertainties provided by the manufacturer. The wind speed derived from the ANE does not appear to be significantly affected by relative winds up to 15 m s$^{-1}$, while the wind speed received from the M600 UAS overestimates the wind speed and appears less reliable (Fig. 4e). The attached payload could cause a miscalculation of the wind speed by the M600 on-board computer, which bases its calculations on the nominal flight behavior of the (payload-free) M600.

The results show that horizontal flight (or relative wind) velocities up to 15 m s$^{-1}$ do not relevantly affect most variables, except for those of the OPC, whose sampling efficiency is apparently reduced at wind speeds > 6 m s$^{-1}$; the internal wind measurement of the M600 seems unreliable with the payload attached. Uncertainties may be due to binning, acceleration and deceleration effects, but also to the normalization to the MoLa time series, which is measured at a distance of up to 130 m and may differ from the FLab data due to spatial inhomogeneity.





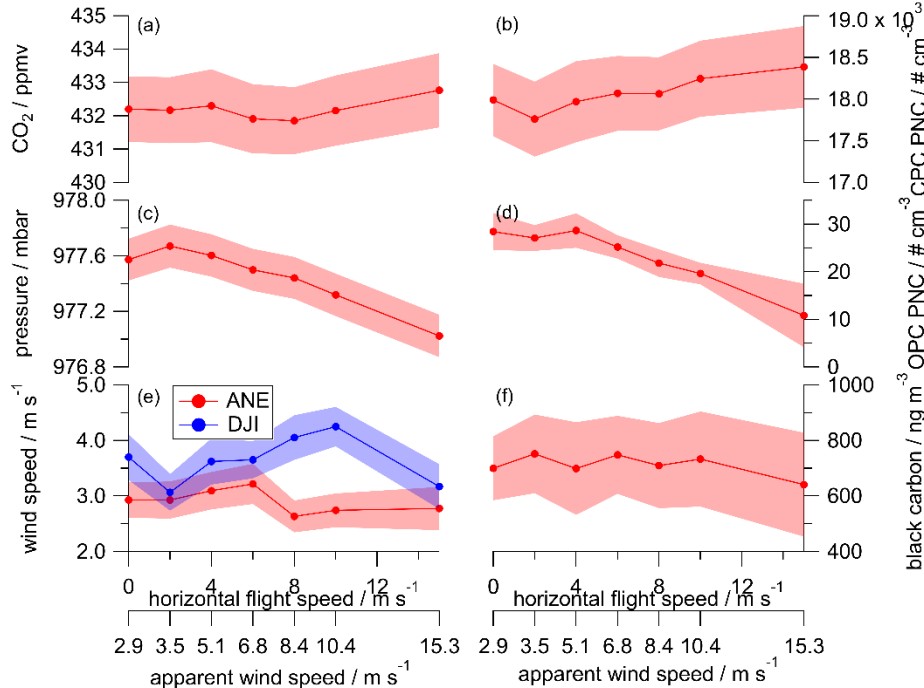

**Figure 4: The variables CO₂ (a) and particle number concentration (PNC) measured with the CPC (b) show no significant dependence on horizontal flight velocity, while pressure (c) and OPC PNC (d) decrease with increasing flight velocity. The ANE wind speed (e, red) and black carbon (f) appear to be completely unaffected, while the wind speed measured by the M600 UAS itself (e, blue) shows an irregular but significant trend. The shaded areas represent the errors calculated as described in Section 3.2. The lines between the markers are for orientation only and are not intended to indicate a relationship.**

## 3.3 Influence of vertical motion

The effect of the vertical velocity of the UAS on the measurements was investigated analogously to the horizontal flights. For this purpose, we performed three flights, F4 to F6, consisting of several vertical profiles up to 120 m AGL with pre-programmed velocities (Table 2). All flights were performed under similar meteorological conditions with no apparent change in air mass (Fig. S7), except that the wind direction changed from east to north before flight F5 and back again after F5, but was stable during F5. Due to the change in wind direction, plumes of CO₂ and small particles were detected at ground level and with MoLa during this flight because the MoLa exhaust was partially sampled.

Following the approach for the horizontal flights, the FLab data of all instruments for each flight were binned in two different ways: with respect to altitude and with respect to vertical velocity, respectively. Binning with respect to altitude results in vertical profiles that contain the average of all measured data, independent of vertical velocity, for each 10 m increment (Figs. 5 and S8; with standard errors shown as uncertainty range). For the analysis of the dependence of the measured variables on the vertical velocity, the measured data were first normalized by the MoLa time series and then by the vertical profiles before



being averaged over the entire altitude range covered for each vertical velocity setting individually, analogous to Sect. 3.2
(Figs. 6 and S9). The total uncertainty includes the uncertainty from the temporal and spatial normalization, from the averaging
of the three flights, and from the statistical uncertainty (Sect. 3.1) for the respective averaging time. Here we assume that the
MoLa measurements are comparable to the FLab measurements independent of the spatial distance, while the actual
uncertainty of this correction increases with increasing vertical distance. A positive vertical velocity indicates ascending
motion and a negative velocity indicates descending motion.

Analogous to the analysis in Section 3.2, the wind speed and direction data collected by the anemometer (ANE in Figs. 5 and
6e) were compared with the data recorded by the UAS (DJI). The wind speed determined by the UAS is almost constant within
$\pm 0.1$ m s$^{-1}$ with respect to the reference wind speed at all altitudes and vertical velocities (Figs. 5a and 6e). In contrast, the
wind speed measured by the anemometer is generally larger and varies with altitude (Fig. 5a), as expected. From this
comparison and the one in Sect. 3.2, we conclude that the UAS-derived wind speed is unreliable with the payload attached and
should not be used for analysis.

From the same figure it can be seen that there are strong differences between the wind speed measurements during ascent and
descent, especially for flights F4 and F6: the ascent-related wind velocities (diamond markers) are greater than those of the
descent (circular markers), measured at the same altitude (Fig. 5a), a behavior observed at all ascent rates (Fig. 6e). This effect
is not only observed for the wind speed measurements, but also for the wind direction measured by the anemometer (Fig. 5b).

There is an average shift of 27° in wind direction measured during descent compared to ascent. This probably indicates a bias
in the wind direction measurement during descent due to the movement into the turbulent downwash volume, which may
contain a general circulation of air caused by the rotation of the propellers (Zheng et al., 2018), while during ascent the UAS
moves into unperturbed air. In our case, the downwash seems to generate a wind field that counteracts the atmospheric wind
in terms of speed and also influences the wind direction. Therefore, the anemometer provides unbiased wind data only during
ascent, and ANE wind data measured during descent should be used with caution.





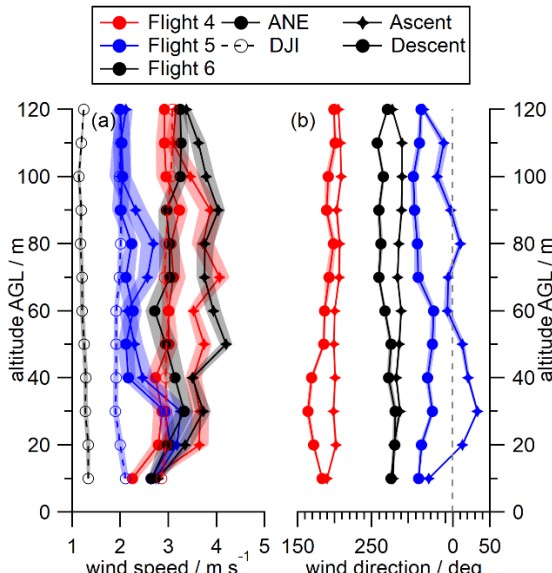

**Figure 5: Vertical profiles of wind speed (a) and wind direction (b) of flights F4, F5 and F6 show the differences between ascent and descent in the ANE data (diamonds and circles on solid lines). Vertical profiles of wind speed obtained from the M600 UAS (DJI, dashed lines) show no altitude dependence. The shaded areas represent the standard error within the 10 m increments.**


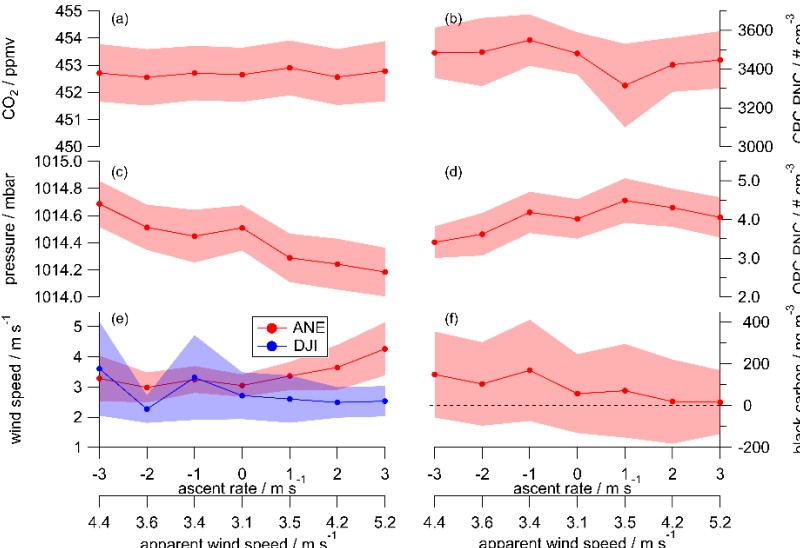

**Figure 6: The variables $CO_2$ (a) and particle number concentration (PNC) measured with the CPC (b) appear to be unaffected by vertical velocity, while pressure (c) and OPC PNC (d) reveal a slight dependence. ANE wind speed (e, red) and black carbon (f) show an apparent trend within the uncertainty range, while the wind speed measured by the M600 UAS itself (e, blue) is not affected by**

**vertical motion. Ascent and descent rates are given as positive and negative ascent rates, respectively. The shaded areas represent the errors as described in the text. The lines between the markers are for orientation only and are not intended to indicate a relationship.**




Most of the other variables do not show a significant relationship with ascent rate (Figs. 6 and S9). No influence of vertical
velocity is found for $CO_2$, $O_3$ and CPC particle number concentration. Patterns within the uncertainty range are found for OPC
(PNC, $PM_{2.5}$ and $PM_{10}$) and for temperature, but are negligible within the given uncertainties. Spatio-temporal overcorrection
seems to cause the non-intuitive but insignificant pattern found for relative humidity (Fig. S9b). In contrast to the relationship
found for horizontal motion (Fig. 4c), the measured pressure decreases with increased ascent rate, but increases with descent
rate. Here, the dynamic pressure dominates over the Bernoulli effect because the pressure sensor is open to the bottom and is
directly exposed to aspirated air masses from below. This small potential bias must be considered when calculating flight
altitude from recorded pressure levels during ascent or descent, although it is within the manufacturer's stated uncertainty.

The measured black carbon mass concentration decreases substantially with increasing vertical velocity during ascent down
to 10% of the descent data. During ascent, the filter with the sampled material is lifted into lower pressure air, presumably
causing evaporation of volatile compounds from the filter and consequently changing the relative amount of light scattering,
which is assumed to be constant for the internal correction. Descending may cause the opposite effect, i.e., condensation of
semi-volatile compounds on the filter, which is colder due to previous measurements at higher altitude. Thus, filter sampling
can be significantly biased when semi-volatile compounds are involved and filters are subjected to changing pressures.
Nevertheless, the black carbon trend is within the extremely large uncertainties of these measurements, which are due to the
low black carbon levels on this day rather than evaporation of the sampled compounds. Therefore, we conclude that the AET
data are not useful for continuous vertical profiling flights in uncontaminated areas, and we do not further consider black
carbon concentrations in absolute terms in this study. Flight patterns that include hovering at the same altitude for several
minutes would be more suitable for filter-based *in situ* measurements.

In summary, the vertical motion of the UAS with velocities in the range of -3 to 3 m s$^{-1}$ does not significantly affect the data
quality of the measured variables, except for AET. Using the same flight pattern for successive profiling flights would result
in better comparability of data between flights. In contrast to previous work investigating vertical velocities $\geq$ 5 m s$^{-1}$ (Brus et
al., 2021b), no hysteresis was found between ascent and descent for the temperature and humidity sensors, indicating that the
instrument sensors can equilibrate sufficiently quickly for the vertical velocity of $\pm$ 3 m s$^{-1}$ used in the given setup.

## 3.4 Development of an optimized vertical profiling flight pattern

Depending on the desired measurement application, very different flight patterns have been used with UAS in the past. In the
literature, circular or systematic mapping horizontal flights are frequently found, which do not include any vertical variation
(Burgués et al., 2021; Grimaccia et al., 2015). In contrast, purely vertical flights allow the measurement of vertical profiles
with separate data for ascent and descent (Andersen et al., 2023; Quinn et al., 2024). In both cases, when large areas need to
be covered, capturing temporal changes can be challenging due to the typically short flight durations. For smaller mapping
areas, such temporal changes can be assessed by repeated flight operations, as shown in flight F1 in Section 3.2. Hover flights
at a fixed location, on the other hand, focus purely on the analysis of temporal variations at high temporal resolution, without





providing information on spatial variations. Hovering also allows slow sensors to equilibrate to ambient conditions and allows longer sampling times to achieve sufficiently low detection limits of the quantities under study (Barbieri et al., 2019; Brus et al., 2021a; Niedek et al., 2023).

Here, we focus on the investigation of vertical profiles with repeated flights over the day to determine the temporal evolution
of the stratification of the lower boundary layer. Based on the results in Section 3.3, a significant influence of the vertical motion of the UAS on the measurement results is not expected for most variables. Wind speed and direction data from the anemometer should only be used when measured during ascent or during hovering. Hover phases near a reference station are highly recommended for all measurement flights to allow frequent comparison of UAS-based measurements with higher-quality instruments. Comparison of data from both platforms helps to detect and correct temporal drifts of UAS-based
instruments exposed to rapidly changing environmental conditions. Low-cost chemical sensors and instruments without sophisticated compensation methods would also benefit from such cross-platform comparisons.

In order to find the optimal flight pattern for vertical profiling, test flights were conducted with a consistent ascent and descent rate of 3 m s$^{-1}$ as a compromise between the statistical uncertainty of the observed variables and the spatio-temporal resolution. A vertical velocity of 3 m s$^{-1}$ allows, for example, four ascents from the ground to 300 m AGL or two ascents to 500 m AGL
within the available flight time of FLab. Furthermore, by combining data recorded during ascent and descent, a total measurement time of 26.4 s per 10 m height bin was obtained for the flight pattern described in Sect. 4.1. Statistical uncertainties are expected to range from 0.08% to 10% depending on the instrument (Table S2). Depending on the research question, the binning can be adjusted, e.g., to improve the measurement statistics.

Occasionally, we observed strong rolling motion of the FLab during descent at low horizontal wind speeds ≤ 3 m s$^{-1}$, resulting
from the UAS entering its own downwash regime. This uncontrolled flight behavior can be avoided by adding a horizontal velocity component of up to 2 m s$^{-1}$ (see Sect. 3.2), directed into the wind to the vertical motion. This causes the UAS to tilt and shifts the downwash regime from below to the side opposite to the direction of flight. As a result, this diagonal rather than straight vertical flight pattern reduces the UAS oscillation without affecting data quality.

This results in a flight pattern that is adapted to the wind speed: straight vertical ascending for wind speeds > 3 m s$^{-1}$ and
ascending with a horizontal component for lower wind speeds. To follow this suggested flight pattern, the pilot must take into account possible changes in wind direction with altitude when planning the flight.

## 4 Applications of FLab

### 4.1 Evolution of the stratification of the lower troposphere

To evaluate and demonstrate the FLab's ability to capture changes in the vertical distribution of aerosols and trace gases during
the day, eight hours of hourly vertical profiling flights were performed. For this experiment, we chose the same measurement site near Ockenheim as described in Section 3, and measured in combination with MoLa (serving as a ground-based reference) on 5 June 2023, between 8:00 and 17:00 (all times are local time). We launched the FLab eight times (once per hour) for



vertical flights up to 300 m AGL with a vertical velocity of ± 3 m s$^{-1}$ without a horizontal component (see Sect. 3.4). The battery capacity of the UAS limited the flight duration to 15 minutes. Each flight consisted of four ascent/descent cycles and

a hover period of 10 s each time the FLab reached the altitude of the MoLa inlet (6 m AGL). The third ascent of the flight at 15:15 was aborted at 100 m AGL due to sudden gusts; a fourth ascent to 300 m AGL was performed. Between the flights, the CBO's measurement cell was covered with a protective cover while the FLab was parked on the landing site without ventilation or shade, in contrast to more recent field deployments.

The meteorological conditions at the measurement site for each day are summarized in Fig. S10; conditions were sunny with

moderate winds.

The recorded FLab instrument data were binned into 10 m increments for all ascents and descents of each flight, and the mean and standard error for the entire flight were calculated for each altitude bin (top panels in the subfigures of Figs. 7 and S11). If the standard error was smaller than the statistically expected uncertainty from Section 3.1, the statistical uncertainty was used instead. FLab and MoLa data can be compared when FLab was hovering near the MoLa inlet below 10 m AGL (bottom

panels in the subfigures of Figs. 7 and S11 with 1-min averages of the MoLa data).

Over the day, the values of relative humidity and ambient temperature in the vertical profiles mainly follow the typical daily trends with increasing temperature until mid-afternoon and correspondingly decreasing relative humidity, analogous to the observations with MoLa (Figs. S10 and S11). While the relative humidity values are rather constant over the whole altitude range covered, the ambient temperatures show a clear decrease with altitude. Wind speed and direction remain constant over

the day at ground level, but show a complex behavior in the profiles, with a general increase in wind speed with altitude, especially in the lowermost fraction of the altitude range (Fig. S11d). In addition, the variation in wind direction is most pronounced in the lower part of the profiles (Fig. S11e).

At ground level, the mixing ratio of $O_3$ increases continuously until 15:00 as a consequence of photochemical processes. $CO_2$ peaks at 9:00 before decreasing by 7.7 ppmv by early afternoon due to increased dilution in the developing mixing layer and

does not indicate any specific emission sources in the vicinity (Oliveira et al., 2007). Both, $O_3$ and $CO_2$ mixing ratios, increase slightly with altitude during the first flight in the morning, but for both trace gases the vertical gradient and the vertical inhomogeneity decrease thereafter as the entrainment zone breaks up and the convective mixing layer develops (Kotthaus et al., 2023). $O_3$ appears to be well mixed up to 300 m AGL from 10:30 and develops according to its diurnal cycle (Law et al., 2008; Neu et al., 1994), however, $CO_2$ mixing ratios measured above 100 m decrease by 2 ppmv compared to the ground,

which could be due to polluted air masses from regional sources that were not transported above 100 m AGL in the short time. At ground level, we measured enhanced $PM_1$ and $PM_{2.5}$ levels by up to 50% between 9:00 and 12:15 with the OPC in MoLa, and a further increase around 16:00; the development of the time series of the FLab OPC particle number concentration (i.e., for $d_{opt} > 350$ nm) and the corresponding $PM_{2.5}$ were very similar (Fig. S11g). The total particle number concentration measured with the CPC doubled during the day and had two broad maxima between 10:00 and 11:00 and 13:00 and 15:00, in addition

to short-term variations. Advection from different nearby aerosol emission sources is an unlikely reason for this behavior, since the wind direction and speed at the ground remained rather constant throughout the day and did not show any changes



in parallel with these trends. In general, the particle-related quantities determined with FLab while hovering near the ground are in agreement with the corresponding values measured with MoLa. However, the vertical profiles of the particle number concentrations measured with the CPC and OPC show strong differences between the two variables due to the different particle

size ranges measured with the two instruments.

OPC PNC generally decrease with altitude (Fig. 7c), although the gradient varies throughout the day. The strongest gradients are found at 13:30 and 16:30, when the highest concentrations were measured near the ground. In contrast, an almost constant background concentration of 7 # $cm^{-3}$ was observed > 200 m AGL during all flights of the day. The observed gradients are consistent with the assumption that particles in the OPC size range are generated at or near the ground and successively

transported to higher altitudes.

In contrast, the vertical distribution of the PNC of small aerosol particles (measured with the CPC) shows several different gradients and the variation of PNC is rather dominated by the different concentrations measured at ground level with no consistent vertical profile (Fig. 7d). However, the CPC PNC increases by an average of $380 \pm 240$ # $cm^{-3}$ in the lowest 50 m AGL. This gradient could be due to inhomogeneous source distributions, similar to those for $CO_2$, and to particle

deposition at ground level. For flights at 9:30, 13:30, and 14:30, strong structures are observed around 150 m AGL, which may indicate stratification not detected by any other instrument. This underscores the importance of measuring different types of variables simultaneously, from meteorological to gas and aerosol particle ones.





**Figure 7: Vertical profiles of O₃ (a) and CO₂ (b) mixing ratios are dominated by the diurnal cycle of the mixing layer and the incoming radiation and show the strongest changes until 12:30. Vertical profiles of the OPC PNC (c) show a successive vertical uplift of particles ($d_{opt}$ > 350 nm) around noon and in the afternoon. Vertical profiles of the CPC PNC (d) appear to be strongly influenced by the ground level PNC, but show inhomogeneous vertical distributions depending on the individual flight. The shaded areas represent the standard error or the statistical uncertainty from Section 3.1, whichever is greater. The bottom panel of each subfigure shows the direct comparison of the FLab (blue) and MoLa (gray) data while FLab was hovering near the MoLa inlet.**

During this field experiment, black carbon mass concentrations did not show any variation with altitude above the noise level, however, the results did reveal some instrumental characteristics that are important for future field operations. The aethalometer measured in single-spot mode for the first few hours and then automatically switched to dual-spot mode (see Sect. 2.2), apparently when the filter loading reached a certain level (Fig. S12). This switch was associated with a sharp increase in the noise level of the measurements (Fig. S12) due to the reduction in sample flow per spot. In addition, for both measurement modes, there were striking differences between the flight and no-flight concentration levels reported by the



instrument, with much lower, sometimes negative, concentrations during the no-flight periods. We hypothesize that the instrument heated up during periods of static sun exposure between flights, which may have caused evaporation of particulate material from the filter, resulting in negative mass concentration values. For unknown reasons, these differences were more

pronounced in the dual-spot mode. Since the standard deviations were also greatly increased in this mode compared to the single-spot mode, we decided to use only the single-spot mode in future applications.

## 4.2 Bridging the gap between ground-based and aircraft-based measurements

Co-located measurements on four different platforms were used to further evaluate the performance and usefulness of the FLab measurements in the lower troposphere. As part of the BISTUM23 measurement campaign in the Swabian Jura in August

2023, we operated MoLa as a ground-based reference station and FLab for vertical profiling up to 500 m AGL. On 10 August 2023, the research aircraft HALO (High Altitude LOng range) orbited the measurement site at 500 m AGL and subsequently ascended to several thousand meters on its way to another research target (Fig. S13). The on-board HALO instruments collected, among others, meteorological, $O_3$, and $PM_{2.5}$ data (Dragoneas et al., 2022; Giez et al., 2022). Following the flyby, a balloon-borne Vaisala RS41-SG radiosonde was launched, reaching 2000 m AGL within 7 min and providing meteorological

data from ground level to the stratosphere.

Figure 8 shows a comparison of variables collected on board the different measurement platforms during and shortly after this flyby. To allow direct comparison of variables from different platforms, the data are not normalized for temperature or pressure at each measurement location. This comparison shows generally good agreement for all variables across all platforms. Below 500 m AGL vertical profiles were obtained with the radiosonde and FLab. For most variables the trends in this altitude range

are consistent; however, for relative humidity an offset of ~10% and for wind speed reduced values of a few meters per second were measured with the radiosonde. These discrepancies between FLab and radiosonde data are discussed in detail in Section C of the Supplement.

Comparison of the radiosonde and HALO data reveals reasonable discrepancies for all available variables, which may be due to the unavoidable temporal and spatial separation of the measurements on the two platforms in the respective altitudes. Similar

variability for measurements with similar temporal distance was observed with the MoLa ceilometer, which detected slight shifts in the backscatter signal profiles (e.g., in the location of the maxima around 1000 m and 1300 m AGL) between the time of the HALO flyby and that of the radiosonde launch (light and dark green traces in Fig. 8, respectively).

Although a direct comparison of HALO and FLab data is not possible due to the lack of overlap in the altitude ranges covered, a reasonable agreement of all variables in the link between the respective profiles at 500 m AGL is observed between the two

platforms. This example shows that FLab is able to reliably bridge observations from ground level to 500 m AGL, an altitude range that is typically not accessible with research aircrafts.



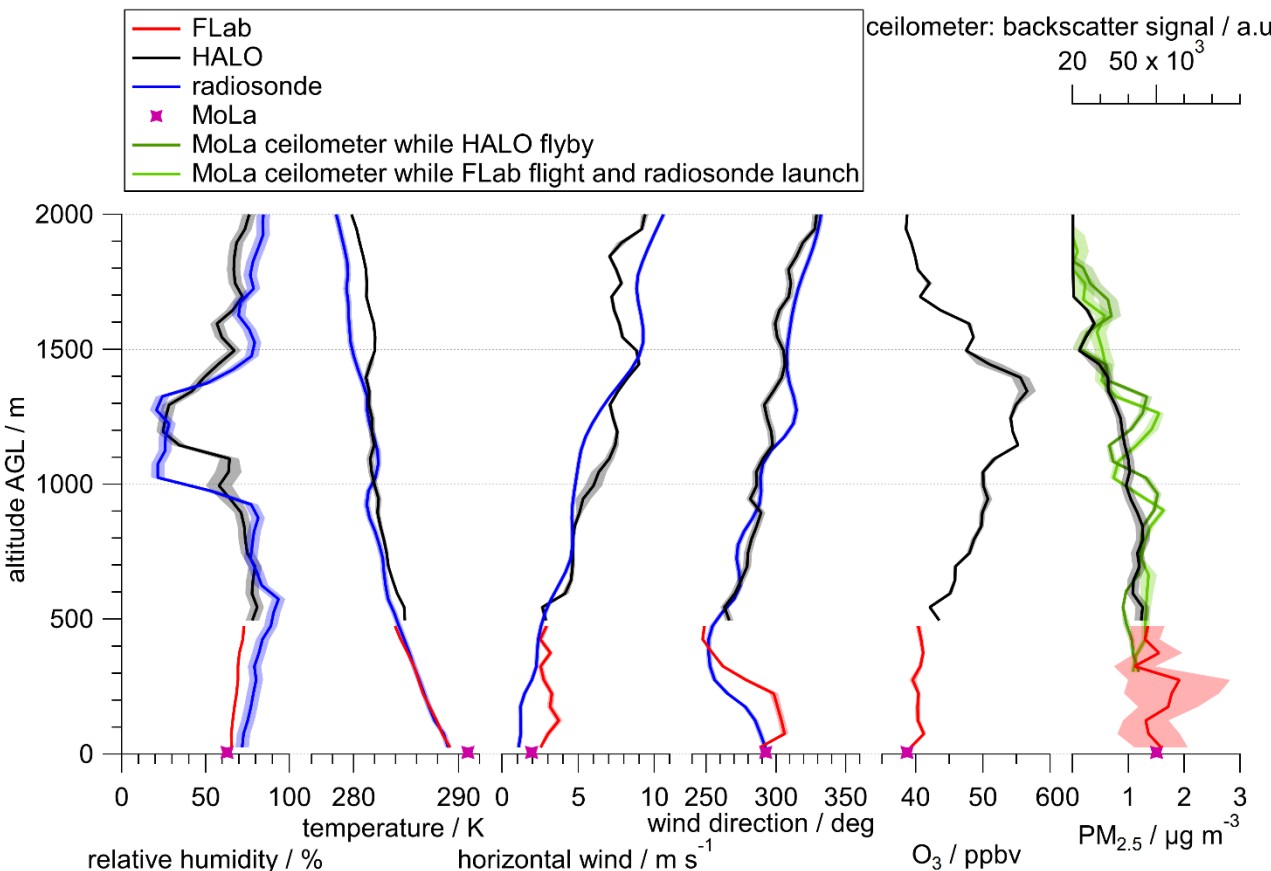

**Figure 8: Vertical profiles of meteorological variables, O₃ and PM₂.₅ show good agreement between different platforms: MoLa (pink and green), FLab (red), radiosonde (blue) and HALO (black). The shaded areas show the respective uncertainty ranges and are derived from the instrumental uncertainty and the standard error of each variable within the 50 m increments.**

## 5 Summary

We have developed and characterized a research UAS (unmanned aircraft system) for atmospheric composition measurements based on a commercial hexacopter (Matrice 600, DJI Ltd.): the Flying Laboratory FLab. FLab was designed as a flexible platform to simultaneously measure aerosol particles, trace gases, and meteorological variables to study the vertical structure of the lowermost atmosphere and the distribution of pollutants in the lower troposphere.

The research UAS was equipped with lightweight and high-quality instruments to measure a wide range of relevant variables characteristic of many types of anthropogenic and natural emissions, such as particle number and black carbon mass concentrations, particle size distributions (from which $PM_1$ and $PM_{2.5}$ concentrations can be calculated), $CO_2$ and $O_3$ mixing ratios, and relevant meteorological variables: air pressure, temperature, relative humidity, and wind direction and speed. An elevated mounting position for the anemometer was identified to measure the ambient wind largely unaffected by rotor





downwash. In addition, we developed a dedicated infrastructure for power and data management for the instruments, as well as hardware and software for real-time ground-based monitoring of the in-situ measurements aboard FLab. In the field, this software can help detect plumes or dangerous flight conditions such as sudden gusts or precipitation.

The research UAS was characterized both on the ground and in flight to evaluate different vertical flight patterns. In laboratory experiments, the measurement uncertainty as a function of the averaging interval was determined for all measured variables, and an optimal averaging time of at least 30 s was determined for our applications to achieve uncertainties of $< 10\%$. Furthermore, we investigated the influence of horizontal and vertical motion with different flight velocities ranging from $0 \text{ m s}^{-1}$ to $15 \text{ m s}^{-1}$ and $0 \text{ m s}^{-1}$ to $3 \text{ m s}^{-1}$, respectively, on the measurement results. From the horizontal flights, it was found

that the horizontal motion slightly affected the temperature and pressure measurements (for velocities $> 2 \text{ m s}^{-1}$); however, this deviation was still within the uncertainties given by the manufacturer of the respective instrument. These experiments also showed a significant decrease in the sampling efficiency of the OPC for wind speeds $> 6 \text{ m s}^{-1}$. Vertical motion of the FLab was found to affect the optical particle counter results in the original setup, which was then modified with an external pump to stabilize the sample flow and a modified inlet to reduce the impact of vertical air advection on the inlet and outlet. Black

carbon measurements turned out to be impractical for continuous vertical profiling flights under unpolluted conditions. We found that straight vertical ascents/descents in windy conditions and diagonal vertical ascents/descents in calm conditions were the safest flight patterns for vertical profiling.

The application of FLab in hourly profiling measurements demonstrated that this approach provides valuable information on the vertical structure of the lower boundary layer and its evolution during the day. Comparison of data measured on board

different platforms revealed that FLab is able to reliably bridge the altitude range between ground-based measurements and low-flying research aircrafts. Co-located measurements with FLab and the mobile aerosol laboratory MoLa show that continuous ground-based measurements can be successfully complemented by the UAS to add information about the vertical distribution of the measured variables. While MoLa is able to measure the temporal evolution, diurnal cycles or plumes of pollutants at ground level, it cannot analyze vertical inhomogeneity or transport processes up to higher altitudes. In combination

with the FLab, we were able to detect vertical inhomogeneities that can occur with the development of the turbulent boundary layer. Thus, the combination of MoLa and FLab provides high quality ground-based measurements while allowing an assessment of their representativeness for the lower boundary layer.

*Author contributions:* LM analyzed the data and drafted the manuscript. TB and PS built the research UAS in close
collaboration with FD, FF and LM. LM programmed the monitoring software and performed the flights together with TB and PS, while FD and FF performed the measurements with MoLa. LV provided and analyzed the radiosonde data. LM, FF and FD discussed the data processing and the presented results. All co-authors commented on the manuscript.

*Competing interests:* The authors declare no conflict of interest.



*Acknowledgements:* This work was supported by internal funding from the Max Planck Society. LM, LV and PS are funded

by the Deutsche Forschungsgemeinschaft (DFG, German Research Foundation) – TRR 301 "TPChange"– Project-ID

428312742. The authors thank the following participants of the PHILEAS project for providing data from the HALO flight:

Andreas Zahn (Karlsruhe Institute of Technology) for the FAIRO $O_3$ data, Andreas Giez (German Aerospace Center) for the

BAHAMAS meteorological data, and Franziska Köllner (Johannes Gutenberg University Mainz/Max Planck Institute for

Chemistry) and Philipp Brauner (Max Planck Institute for Chemistry) for the sky-OPC data.

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
