# Peer review of "The Flying Laboratory FLab: Development and application of a UAS to measure aerosol particles and trace gases in the lower troposphere"

_EGUsphere, 2024_

## Author Comment (AC1)

**RC1: 'Comment on egusphere-2024-3566', Anonymous Referee #1, 03 Jan 2025  reply**

This manuscript introduces the Flying Laboratory (FLab), a hexacopter equipped with six instruments to measure aerosol properties, trace gases, and meteorological parameters in real time with high temporal resolution. Designed to monitor urban and semi-urban environments near aerosol sources, FLab bridges the gap between ground-based and aircraft profiling. During field experiments, it performed hourly vertical profiling flights up to 300 m for eight hours, capturing lower convective boundary layer dynamics and afternoon vertical particle transport. This study details the FLab's technical setup, measurement characterization, and optimized flight patterns, showcasing its effectiveness in environmental monitoring. Overall, the manuscript is well-written. The results are interesting and valuable to the literature. I have some concerns especially for the measurement of wind and the impacts of flight speed on the pollutant measurement.

➔ Thank you for this positive general comment. We reply to the concerns regarding the wind measurements and the impact of the flight speed further below.

Description of the UAS: Please provide additional details about the UAS, including its maximum flight speed, ascent and descent rates, and vertical and horizontal operational ranges. This information will help contextualize the measurements.

➔ We adopted the reviewer's suggestion and included this information in the main text in lines 99-103:

"The maximum flight velocity is 18 m s$^{-1}$ with a maximum ascent and descent rate of 5 m s$^{-1}$ and 3 m s$^{-1}$, respectively. FLab is capable of operating at altitudes up to 2500 m AMSL (above mean sea level; higher with an optional high-altitude propeller set) and at distances up to 5 km; however, in our measurements legal limitations restrict its operation to altitudes of up to 500 m AGL (above ground level) and horizontally up to distances where the aircraft is still visible with the bare eye."

Line 150: Include the full names of the instruments mentioned in the main text to ensure clarity and accessibility for readers unfamiliar with the abbreviations.

➔ We adopted the reviewer's suggestion and included the full names of the instruments in the main text of the manuscript at the first location the instruments are mentioned. In addition, the model and the manufacturer of each instrument are provided in the footnotes to Table 1 in the main text.

Lines 155-156: The anemometer was installed 110 cm above the UAS. Please discuss how UAS vibrations during flight might influence wind measurements. Have any corrections, such as those derived from wind tunnel experiments, been implemented?

➔ Video graphical analysis revealed a vibration frequency of the anemometer setup of 10 ± 1 Hz with an amplitude of up to 2 cm. For 1-second data, the influences of these vibrations largely (>95%) cancel each other out, resulting in a potential bias of the measured wind speed of up to 0.02 m s$^{-1}$, well within the uncertainty of the anemometer measurements of 0.1 m s$^{-1}$ (Anemoment, 2021). For longer averaging times, the uncertainty of the wind measurement due to vibration is reduced even more and therefore does not require further corrections.
We have added this information to the main text in lines 250-253:

"The anemometer mount has a vibration frequency of 10 ± 1 Hz with a maximum amplitude of 2 cm, as determined during the flights using video graphical analysis. Therefore, for the 1-

second anemometer data, the effects of the setup vibration on the wind measurement largely cancel out, resulting in a potential bias of less than 0.02 m s$^{-1}$, well within the instrument's uncertainty of 0.1 m s$^{-1}$ given by the manufacturer (Anemoment, 2021)."

Lines 155-156: Explain whether solar heating could affect the relative humidity and temperature measurements obtained by the anemometer.

➔ Indeed, solar heating could affect the relative humidity (measured by a relative humidity sensor) and temperature (derived from speed of sound and relative humidity) measurements by the anemometer. However, especially during flights, with substantial air movement around the instrument, the relative humidity sensor is well ventilated. The opening at the bottom of the anemometer case provides additional ventilation of the sensor, while the remaining case still shades the sensor from direct sunlight. We have added a statement about these facts in lines 163-166:

"The downward-facing opening in the housing allows for rapid air exchange at the humidity sensor while the sensor is shielded from direct solar radiation, minimizing the effect of solar heating on the humidity measurement. The temperature is determined from relative humidity and speed of sound and consequently not strongly affected by solar heating. "

Line 161 CPC measurement: Provide further details on why the CPC and OZN did not require in-flight adjustments apart from calibration. Additionally, clarify the CPC inlet's placement and assess the potential impact of UAS-induced airflow on the measurements.

➔ The concentrations of gases and small particles are expected to quickly equilibrate around the payload below the FLab due to the turbulent air motion caused by the downwash. Particle number concentrations measured with the CPC are massively dominated by ultrafine particles, which are not strongly affected by impaction, sedimentation losses, or sampling effects. Therefore, for the CPC as well as the ozone measurements short inlet lines to the side of the instrument cage provide sufficiently reliable sample transport to the instruments and representative concentrations, resulting in measurements that do not require corrections for sampling and transport influences, except for a short sampling delay of 2 s for the CPC measurements due to the transport time. This sampling delay is only important when considering 1 s data; for longer averaging times, its effect becomes negligible (see also reply to comment below).
We modified lines 170-176 to make this clearer:

"The CPC and OZN did not require adjustments for in-flight operation other than calibration. This is because concentrations of small particles (which dominate the CPC measurements) and gases quickly equilibrate around the FLab instrument cage due to the turbulent air motion, induced by the downwash. Since sampling and transport losses (other than diffusion) are negligible for both variables, short inlet lines to the side of the instrument cage provide representative sampling and reliable sample transport to the instruments. A short sampling delay of 2 s (derived theoretically for the design of the inlet system and experimentally validated) was found for the CPC and corrected accordingly in the data analysis."

Lines 218-219: Consider rephrasing to "If there is no difference in wind speeds, it indicates 100% agreement between the FLab and MoLa anemometers" for improved clarity. The current phrasing

could be misinterpreted as suggesting no difference exists between the two, which contradicts Figure 2.

➜ We adopted the reviewer's suggestion in lines 233-235:

"For each anemometer position, the difference between the averaged wind speeds measured on the UAS and on MoLa was calculated for both measurement situations, i.e., with the propellers operating and with the propellers turned off. If there is no difference in wind speeds, it indicates 100% agreement between the FLab and MoLa anemometers."

Figure 2: Emphasize in the text that these results correspond to maximum propeller rotation rates. Propeller-induced disturbances may be smaller during hovering

➜ We adopted the reviewer's suggestion and emphasized the reason of using the maximum propeller speed in lines 235-238:

"In this experiment, the propellers were operated at full speed to determine the maximum spatial extent and maximum influence of the downwash, which depends on the propeller's load and therefore may be smaller in-flight depending on the respective flight maneuver and wind conditions."

Lines 224-226: Clarify whether "lower mounting height" refers to the UAS or the anemometer. The current phrasing could lead to confusion and appear inconsistent with Figure 2.

➜ We clarified that the "lower mounting height" refers to the anemometer in this sentence.

Lines 230-235: Again, have experiments, such as wind tunnel tests, been performed to assess the effects of UAS tilting on anemometer wind measurements? If not, consider providing recommendations for future research to address this potential limitation.

➜ Thank you for this comment. The TriSonica Mini anemometer, which is mounted on FLab, was especially designed for applications as on UAS, where the device might be tilted during horizontal movement or when the UAS balances against the wind. This anemometer is specified by the manufacturer to provide 3D wind information (horizontal and vertical component) up to an angle of incidence of 15° in a coordinate system relative to the anemometer. By correcting for the yaw, pitch and roll angle provided by the UAS, this measured 3D wind is transformed into a terrestrial coordinate system and thereby corrected for tilting of the system, as described in Sect. 2.2.1.
During flights with horizontal flight velocities up to 15 m s$^{-1}$, FLab has a maximum tilting angle of 11°, therefore we expect the anemometer to measure the wind speed correctly. Indeed, horizontal flight in-field characterization did not show any significant influence of different angles of incidence (during flights with different horizontal velocities) after transformation into the terrestrial coordinate system, as shown in Sect. 3.2. To make this clearer, we added the following statement to the main text of the manuscript in lines 253-256:

"According to the manufacturer, the anemometer is capable of measuring 3D wind up to angles of incidence of 15°, and in our characterization measurements (Section 3.2), we do not observe any significant influence of the horizontal flight velocity (up to 15 m s$^{-1}$, which relates to a maximum tilt angle of 11°) on determined wind speed."

Lines 280-281: 370 nm corresponds to the light absorption of brown carbon. It may not be appropriate to state that black carbon concentration was determined from the 370 nm measurement. Consider rephrasing to "light-absorbing species."

➔ We adopted the reviewer's suggestion.

Lines 302-303: Include a brief description of the Allan variance to help readers unfamiliar with this analytical method.

➔ Thank you for pointing out that the Allan variance may be unfamiliar to some readers. We rephrased and extended the description of our analysis in the main text by explaining how to differentiate between statistical noise and influences of temporal trends in Allan diagrams as used in Fig. S3 (lines 325-333):

„To separate the statistical noise in the time series from the influence of the signal's trends over time, we calculated the Allan variance. The Allan variance is the statistical variance of the differences between adjacent data points as a function of the averaging time. When plotting the Allan variance versus the corresponding averaging times (Allan variance plot, e.g., Fig. S3), typically a decrease of the Allan variance with increasing averaging time is observed due to decreasing influence of statistical noise. At some point, the Allan variance reaches a minimum and thereafter starts to increase again. This is when the influence of temporal trends becomes more important than statistical differences between adjacent data points. The minimum in the Allan plot separates averaging times where statistical noise dominates from those where temporal trends dominate the variability in adjacent data points (Werle et al., 1993)."

Lines 333-336: Provide guidance on optimal averaging times for different instruments and discuss whether these recommendations might vary under different flight conditions, such as varying speeds or wind intensities.

➔ Generally, the more robust results are obtained with longer averaging times, which consequently reduces the temporal resolution of the results. However, in flight planning, not only the averaging time has to be taken into account, but the achievable spatiotemporal resolution is also determined by the flight pattern (e.g., flight velocity) and atmospheric conditions (e.g., wind speed). In the end, the desired temporal and spatial resolution, in conjunction with the total available flight time and the choice of the flight pattern, determines the maximum possible averaging time, and none of these can be considered separately. We added a statement about this to lines 363-365:

"Since the most suitable averaging time depends on the scientific goal and which levels of uncertainties are acceptable, as well as on desired flight pattern and atmospheric conditions, no general recommendation can be given. Nevertheless, this study can help to estimate expected uncertainties and herewith support mission planning."

Lines 362-363: Rephrase or include an example to clarify the method of normalization used in the study.

➔ We found that the term "normalization" in our original manuscript is misleading. We rephrased the paragraph that describes the temporal and spatial correction method in lines 395-411 for clarification:

"In order to investigate potential effects of flight velocity on the measurements, the

acquired FLab data need to be corrected for variations that do not originate from flight velocity influences but from temporal and spatial trends in the probed air. First, the time series of each variable of the FLab data were corrected for temporal variations in the reference data (measured by MoLa; these data were smoothed by calculating the 15 s rolling mean in order to minimize noise). The temporal correction is done by multiplication of each FLab data point with the ratio of the respective data point in the 15 s rolling mean time series of the MoLa data to the first data point in this time series. For this we assume that temporal variations are the same at all measurement locations due to the (assumed) homogeneous atmospheric distribution over this range. Second, a correction for the average spatial structure (i.e., average horizontal trend or, as in Section 3.3, vertical profile) was applied to the FLab data for each variable. For this purpose, average horizontal or vertical profiles (20 m, respectively 10 m bin size) were calculated from the FLab time series after correcting them for the temporal trends (see Fig. S7b). Then, the data in the FLab time series were corrected for this average spatial distribution by multiplication of each data point with the ratio of the value in the average profile for the respective position to that for the reference position (i.e., the 0 m mark; Fig. S7c). Finally, the thusly corrected FLab data were rebinned according to flight velocity by calculating for each variable the mean over all flights and all positions at a given velocity (Fig. S7f). Combined uncertainties from temporal and spatial correction, individual instruments, and flight-to-flight variability were estimated and are shown as error ranges in the respective figures. Figures 4 and S8 show the resulting dependence of different variables on flight velocity and apparent wind speed, respectively. The apparent wind speed is calculated by adding the vectors of flight velocity and ambient (horizontal) wind."

Additionally, graphical clarification of the different steps of the correction method is shown in the newly added Fig. S7:

[Figure]

**Figure S7**:  The average temperature profile (20 m bins) is calculated from the uncorrected data from the horizontal flights from Section 3.2 (shown in panel a). The underlying temperature data from (a) are plotted depending on the horizontal flight velocity for each flight in (b), i.e., (b) shows the temperature dependence on flight speed if temporal and spatial variations are not regarded. (c, d) shows the same, using data corrected for temporal variation in ambient temperature; while in (e,f) the data were additionally corrected for spatial variability (see Section 3.2); (f) is the final corrected dataset.

Lines 365-368: This could be due to the fact that the flights were conducted under relatively homogeneous conditions as the authors mentioned in line 277. However, how does the UAS perform in capturing pollutant concentration variations at different flight speeds over heterogeneous land surfaces?

➔ The experiments in Sections 3.2 and 3.3 were designed to study how flight speed affects the measurement of variables. Therefore, it was important to perform these experiments under homogeneous conditions to minimize any influences that weren't measured at the reference station, MoLa. However, a single event during one of the flights (Flight 3) demonstrates that the instruments also perform well in capturing short-term changes of pollutant concentrations at different flights speeds. During flight 3 of the horizontal characterization measurements, the wind direction changed, and the MoLa exhaust plume polluted a short section of the flight track, as visible in the average in Fig. S5h and shown for the underlying 1-second data of the particle number concentration in the new Fig. S8 (see below). The UAS was operating at different horizontal flight velocities during the different legs of this flight, and the plume was clearly detected from the third leg on, independent of the flight velocity. Particle number concentration increased by more than 5000 particles cm$^{-3}$ at various locations (depending on the slightly changing wind direction) between the -20 m and 0 m-mark during flight 3, illustrating FLab's capability to capture local inhomogeneities of pollutant concentrations independent of flight velocity.

For clarity, this figure was added to the supplement and an explanation was added in lines 391-394 of the manuscript:

"Figures S5h and S6 highlight the Flab's capability of resolving such small-scale inhomogeneities like the MoLa exhaust plume on a 1-second-time scale within a few tens of meters and independent of horizontal flight velocity, resulting in enhanced CPC PNC in the averaged horizontal flight tracks in this example."

[Figure]

**Figure S6**: The CPC on FLab measures increased particle number concentrations from local plumes between the -30 m and 0 m-mark of the horizontal flight tracks during Flight 3.

We found that the CPC has a sampling delay of 2 seconds (predominantly due the slow flow rate through its measurement cell), which only becomes apparent when considering 1s-data. Therefore, we corrected for this and reproduced the whole analysis where measurements with the CPC are concerned; since all presented analyses relied on longer averaging times, most changes were negligible:
Figures 3, 4b, 6b and 7d don't show differences to the previous analysis.
Figure S5h was updated and shows a slightly narrower plume with slightly increased CPC particle number concentrations for the -20 m and 0 m mark, instead of the -40 to 20 m mark as before, especially for flight 3.
Figure S6 was added to the revised Supplement.
Figure S8 (now S10) was updated without significant changes.

We also reassessed all other instruments with regards to this, but found that no sampling delay was observed or to be expected from theory; therefore, no further changes were necessary.

Figure 4e: Where did the authors obtain the DJI UAS wind speed data? Have the authors compared the DJI UAS wind speed data with anemometer measurements during hovering? Additionally, address the uncertainty associated with the anemometer and whether environmental factors such as temperature, humidity, and pressure could influence its performance.

➔ The DJI UAS data is an output variable from the on-board computer of the UAS Matrice 600. The DJI UAS wind speed reflects the horizontal wind speed, which is calculated from the drones' GPS data and the thrust force from the individual propellers, using the factory calibration for the UAS. The thrust forces, however, also depend on the weight and center of gravity of the UAS and differ when changes are made to these variables. Therefore, a new calibration is recommended to take these changes into account when, e.g., a payload is attached or modified. Ambient conditions (e.g., temperature, humidity) may influence the propellers' thrust force; however typically only insignificantly (Abichandani et al., 2020). The uncertainty of the DJI wind speed is unknown. A comparison of this DJI wind speed with alternative measurements has not been published in the literature, yet.

The information on how the wind speed is determined by the M600 is given in Sect. 2.2.1; for clarity, we briefly added this now also in the discussion of Figures 5 and 6, which also highlights that the payload presumably influences the wind speed determined by the M600 ("DJI"):

Lines 421-425:
"The ambient wind speed derived from the ANE does not appear to be significantly affected by relative winds up to 15 m s$^{-1}$. In contrast, the wind speed determined by the M600 UAS, which is based on the GPS data and the rotors' thrust force, overestimates the wind speed and appears to lack reliability (Fig. 4e). The attached payload could lead to a miscalculation of the wind speed by the M600 on-board computer, which bases its calculations on the nominal flight behavior of the (payload-free) M600."

➔ A hover flight comparison of the wind speed measured with the anemometer ("ANE") and derived from the UAS position and rotors' thrust ("DJI") is depicted in Figs. 4e and 6e, where the 0 m s$^{-1}$ horizontal resp. vertical flight velocity represents hover flight conditions. As can be seen from these figures, the wind speeds agreed within the uncertainty for the vertical flights, while they were slightly higher for DJI compared to ANE for the horizontal flights.

Lines 410-415: Clarify that the anemometer measured horizontal wind in this section.

➔ We included this information in this section.

Lines 460-462: While the temperature and humidity sensors seem to equilibrate quickly at high flight speeds, how do the CPC and pollutant sensors compare? Their response times might be slower, particularly given their placement underneath the UAS. Have the authors tested how these sensors respond to rapid changes in flight conditions, such as transitioning from high-speed ascending flight to hovering? If they respond quickly, the concentrations would likely stabilize during hovering. If not, there could be delays or noticeable overshoots or drops in concentration readings during such transitions.

➔ In previous experiments and in agreement with the instruments' manuals, we found that the CPC and the other pollutant sensors respond within a few seconds to sudden changes in concentrations. Therefore, no further characterization of these instruments, regarding response time, after installation on-board FLab was conducted.

The flight through an already slightly extended engine exhaust plume, which is depicted in Figs. S5h and S6 shows the ability of the CPC to capture sudden pollution concentrations changes within the few seconds of plume crossing. This is stated in the revised manuscript in response to the comment on lines 391-394 (see above).

**References:**

Abichandani, P., Lobo, D., Ford, G., Bucci, D., and Kam, M.: Wind Measurement and Simulation Techniques in Multi-Rotor Small Unmanned Aerial Vehicles, IEEE Access, 8, 54910-54927, https://doi.org/10.1109/access.2020.2977693, 2020.
Anemoment: TriSonica Mini Sensors: User Manual, 2021.

---

## Author Comment (AC2)

**RC2: 'Comment on egusphere-2024-3566', Anonymous Referee #2, 03 Jan 2025  reply**

Overview:

The manuscript outlines the development and testing of a Flying Laboratory Flab using an uncrewed aerial system equipped with various instruments to measure meteorological parameters and in situ trace gases and aerosol particles. The Flab is equipped with an anemometer for measuring wind speed and direction along with temperature, humidity and pressure, CO2 and O3 analyzers, instruments for measuring aerosol particles. The manuscript outlines various tests to determine data uncertainties and optimal flight parameters for data accuracy and precision. The treatment is thorough and the manuscript is well written. I recommend this manuscript for publication.

➔ Thank you for this positive judgement of the manuscript and the constructive comments. Individual comments are answered below.

Minor Comments:

Lines 37-44: In addition to the limitations outlined by the authors, lidar instruments specifically have near-field dead zones and when used from the ground, cause there to be a lack of continuity between ground observations and remote observations. UAS are ideal for bridging this gap.

➔ Thank you for pointing this application case out. We included a brief mentioning of this limitation of lidar applications to the introduction, implicating the additional advantage of using UAS to measure the near-field zones in lines 40-41:

"However, these methods are limited by near-field dead zones close to the instrument, leaving the lowermost part of the atmosphere uncovered."

Line 125 and line 132-134. There are some statements here which are confusing. I believe that all of the language here is used to describe the aerosol instruments but when it states 'an Arduino Uno is used to store the processed instrument data on a common SD card for all instruments' it could be interpreted that all instruments on board are backed up to the SD card. Perhaps it would make more sense to describe that the ozone monitor and the aethalometer only use internal storage (is that the correct interpretation?). Start with the data storage that is independent and then qualify the integrated data storage (and transmittance) for the aerosol instruments as only for those aerosol instruments.

➔ Thank you for the suggestions. Actually, the Arduino indeed was used to store data from all instruments, not only from those which did not store data internally. We revised the text to make this clearer. We also adopted the reviewer's suggestion and now first mention the instruments which store data independently and then describe how all instruments send their data to the Arduino microcontroller in real time (even the ones that store data individually anyway) in lines 126-129:

"The ozone monitor and the aethalometer store their data in independent internal memories. In addition, both instruments as well as all the other instruments transmit measurement data in real-time to Arduino Mega microcontrollers (ATmega 2560)."

The description of the independent internal memories from lines 133-134 was removed.

➜ Additionally, we clarified how the data is transformed (from raw to transmittable data) on the Arduino in lines 129-131:

"Due to the low processing power of the Arduino Mega, three microcontrollers are required in the FLab to receive raw data and to process the partially large output strings to compact data packages."

Line 151: Which instruments had RS232 interfaces which required modification. Just list parenthetically.

➜ Reply: We included this information in lines 156-158, according to the reviewer's suggestion.

Line 371: requires a space between parentheses and 'which'

➜ Reply: Done, thank you.

Lines 410-415: The statement 'the wind speed determined by the UAS is almost constant within 0.1 ms-1 with respect to the reference wind speed at all altitudes and vertical velocities' implies that there should be some indication of the reference wind speed in Figures 5 and 6 but there is none. What do you mean by reference wind speed? Also the statement 'the UAS-derived wind speed is unreliable with the payload attached' is confusing. Are there observations of UAS-derived winds which are accurate but not so once the payload is attached? Reference performance of only UAS-derived winds with no payload or something to clarify. Or this paragraph needs some statement to agreement with reference wind speeds – the figures that are referenced are only comparing the anemometer and UAS but I don't believe either of them to be considered reference.

➜ Thank you for pointing this out. The sentence was indeed poorly worded and therefore misleading. What was meant was that the DJI wind speed was constant within +/- 0.1m/s, whereas the anemometer onboard Flab (misleadingly termed "reference") did not show this behavior. We rephrased the sentence in lines 456-457 to:

"The wind speed determined by the UAS is almost constant within ± 0.1 m s$^{-1}$ for all altitude levels within each flight (Figs. 5a and 6e)."

As described in Sect. 2.2.1, the attached payload may lead to miscalculation of wind speed as calculated by the UAS since this is derived from the thrust force on the propellers and the GPS data. When a payload is attached to the UAS, the propellers experience an increased thrust force, which results in a biased wind speed calculation. A detailed explanation can be found in Wildmann and Wetz (2022).

To clarify how the UAS wind speed is derived this information was now also added to lines 421-425:

"The ambient wind speed derived from the ANE does not appear to be significantly affected by relative winds up to 15 m s$^{-1}$. In contrast, the wind speed received from the M600 UAS, which is based on the GPS position and the rotors' thrust force, overestimates the wind speed and appears less reliable (Fig. 4e). The attached payload could cause a miscalculation of the wind speed by the M600 on-board computer, which bases its calculations on the nominal flight behavior of the (payload-free) M600."

**References:**

Wildmann, N. and Wetz, T.: Towards vertical wind and turbulent flux estimation with multicopter uncrewed aircraft systems, Atmospheric Measurement Techniques, 15, 5465-5477, https://doi.org/10.5194/amt-15-5465-2022, 2022.